# Understanding Differences in Cancer Survival between Populations: A New Approach and Application to Breast Cancer Survival Differentials between Danish Regions

**DOI:** 10.3390/ijerph16173093

**Published:** 2019-08-26

**Authors:** Marie-Pier Bergeron-Boucher, Jim Oeppen, Niels Vilstrup Holm, Hanne Melgaard Nielsen, Rune Lindahl-Jacobsen, Maarten Jan Wensink

**Affiliations:** 1Interdisciplinary Center on Population Dynamics, University of Southern Denmark, J.B Winsløws Vej 9B, 5000 Odense C, Denmark; 2Department of Oncology, Odense University Hospital, J.B Winsløws Vej 4, 5000 Odense C, Denmark; 3Danish Twin registry, Institute of Public Health, University of Southern Denmark, J.B Winsløws Vej 9B, 5000 Odense C, Denmark; 4Department of Oncology, Aarhus University Hospital, Palle Juul-Jensens Boulevard 99, 8200 Aarhus N, Denmark; 5Department of Epidemiology, Biostatistics and Biodemography, University of Southern Denmark, J.B Winsløws Vej 9B, 5000 Odense C, Denmark

**Keywords:** breast cancer, decomposition, stage at diagnosis, Denmark, regions, survival

## Abstract

Large variations in cancer survival have been recorded between populations, e.g., between countries or between regions in a country. To understand the determinants of cancer survival differentials between populations, researchers have often applied regression analysis. We here propose the use of a non-parametric decomposition method to quantify the exact contribution of specific components to the absolute difference in cancer survival between two populations. Survival differences are here decomposed into the contributions of differences in stage at diagnosis, population age structure, and stage-and-age-specific survival. We demonstrate the method with the example of differences in one-year and five-year breast cancer survival between Denmark’s five regions. Differences in stage at diagnosis explained 45% and 27%, respectively, of the one- and five-year survival differences between Zealand and Central Denmark for patients diagnosed between 2008 and 2010. We find that the introduced decomposition method provides a powerful complementary analysis and has several advantages compared with regression models: No structural or distributional assumptions are required; aggregated data can be used; and the use of absolute differences allows quantification of the survival that could be gained by improving, for example, stage at diagnosis relative to a reference population, thus feeding directly into health policy evaluation.

## 1. Introduction

Large variations in cancer survival have been reported between populations worldwide, including between European and high-income countries [1,2]. For example, cancer survival is generally lower in Denmark than in a comparable country like Sweden [1,3,4]. Cancer survival differences also occur at sub-national levels, such as lower survival for males than for females in the same country [5,6,7], or differences between regions [8]. These variations in cancer survival suggest that the gap could be lessened if low-survival populations could approximate survival from the high-survival populations by, among other factors, improving national healthcare systems [9] or reducing socioeconomic disparities [10].

Potential explanations of the differences between populations include: more adverse stage at diagnosis [11], greater burden of certain risk factors (e.g., smoking) [12], or biological differences [13]. Studies showed that stage at diagnosis can be a key explanation for cancer differences between countries [11,14,15].

Cancer survival and mortality differences between populations can be studied with descriptive statistics and comparison of survival functions [1,14,15]. These studies show the difference in survival between populations but provide limited information on why these differences occur. For instance, a more adverse stage distribution can be observed in a low-survival population relative to a high-survival population, but this provides no quantification. Regression analyses, such as the Cox proportional hazard model or other forms of generalized linear models, have been used to study the relation between cancer survival or mortality and a set of independent variables (e.g., stage at diagnosis) [11,16,17]. However, regression model assumptions do not always hold (e.g., proportional hazards). Additionally, these models often estimate relative differences only (e.g., hazard ratio, relative risk, or odds ratio), while absolute numbers are sometimes preferable. Absolute differences can be useful for quantifying the differences explained by a specific variable, or the potential gains in survival that could be achieved by modifying that variable.

Non-parametric decomposition methods are valued tools in demography, yet less common in public health [18], and can quantify the exact contribution of specific components, such as ages and causes of death, to a (usually) absolute difference between populations in a given measure [19,20,21,22]. Many decomposition methods require no assumptions (structural or distributional) about the data.

We introduce the Kitagawa decomposition method [22], extend its application to cancer research, and present a novel extension of the method to account for the confounding effect of background population (incl. background survival). We decompose the difference between cancer survival probabilities in two populations by their underlying differences in (1) age composition at diagnosis, (2) stage composition at diagnosis, and (3) age–stage-specific survival. Each of these contributions can relate to different issues in health care. The method is illustrated by decomposing differences in female breast cancer survival between Danish regions. Denmark has five administrative regions: The Capital, Zealand, Southern Denmark, Central Denmark, and Northern Denmark. In 2007, Denmark established a new political and administrative scheme in which the healthcare system is run at three levels: the state, the regions, and the municipalities. The national level regulates and supervises health and elderly care. The regions are responsible for hospitals, general practitioners (GPs), and psychiatric care units. The municipalities mainly oversee primary healthcare services, such as health promotion or extramural rehabilitation [23]. Given the important role of GPs, hospitals and their interactions in prompt diagnosis, and the important role of hospitals in treatment, all falling under the regions’ responsibility, differences in cancer survival between Danish regions are an instructive test case for the method.

## 2. Methods

The Kitagawa method quantifies the amount of absolute difference in a crude rate that is due to differences in compositions versus differences in component-specific rates between populations, using multiple standardizations. Extensions and similar techniques have been developed [24,25,26,27]. The Kitagawa decomposition can also be applied to probabilities. Indeed, the Kitagawa decomposition can be applied whenever one variable can be expressed as the product of two others, one of which is, generally, a composition. For example, we can express a crude survival probability (S) as the product of some x-composition $c_{x}$ (e.g., age composition) of a population and the x-specific survival $s_{x}$:(1)$$S=\;\sum_{x}s_{x}{\;c}_{x}$$

To quantify the absolute difference in a crude survival probability between two populations (Y and Z) due to differences in the composition of variable x ($c_{x}$) and differences in x-specific survival probabilities $\left(s_{x}\right)$, the Kitagawa formula reads as:(2)$$S^{Y}-S^{Z}=\Delta \;S=\;\sum_{x}\Delta {\;s}_{x}\;\bar{c_{x}}\;\}\mathrm{survival}\;\mathrm{effect}\;\;+\sum_{x}\Delta {\;c}_{x}\;\bar{s_{x}}\;\}\;X-\mathrm{effect}$$
where the bar over the indicator (probability s or composition c) represents the x-specific averages across populations Y and Z (e.g., $\bar{s_{x}}=\left(s_{x}{}^{Y}+s_{x}{}^{Z}\right)/2$) and Δ is the difference between the two populations (e.g., $\Delta \;s_{x}=s_{x}{}^{Y}-s_{x}{}^{Z}$). If variable x is age, then Equation (2) quantifies:The survival effect: The difference in the crude probability S between two populations that is due to the difference in their age-specific probabilities $\Delta {\;s}_{x}$, i.e., the sum of age-specific survival probabilities differences after multiplying $s_{x}$ for each population by the average age-composition $\bar{c_{x}}$ of these two populations (direct standardization).The age effect (X-effect): The difference in the crude probability S between two populations that is due to the difference in their age-specific composition $\Delta {\;c}_{x}$, i.e., the sum of the age composition differences after multiplying $c_{x}$ for each population by the average age-specific probabilities $\bar{s_{x}}$ of these two populations (indirect standardization).

More than one composition effect can be of interest to understand differences in a rate or probability between two populations. When using any two compositions (x and *i*), the crude survival can be expressed as:(3)$$S=\;\sum_{x}\sum_{i}s_{\mathrm{xi}}{\;c}_{i\left(x\right)}c_{x}=\;\sum_{i}\sum_{x}s_{\mathrm{ix}}{\;c}_{x\left(i\right)}c_{i},$$
where $c_{i\left(x\right)}$ is the i distribution for each x and $c_{x\left(i\right)}$ is the x composition for each i. For example, if x is age and i is stage, then $c_{i\left(x\right)}$ is the stage-composition at age *x*. Kitagawa (1955) [22] provided a way to decompose crude rate differences into two or more composition effects (X and I) and one survival effect:(4)$$\Delta \;S\;=\;\sum_{x}\sum_{i}\Delta {\;s}_{\mathrm{xi}}\;\bar{c_{\mathrm{xi}}}\;\}\mathrm{survival}\;\mathrm{effect}\;\;+\;\sum_{x}\sum_{i}\Delta {\;c}_{x\left(i\right)}\;\bar{c_{i}}\;\bar{s_{\mathrm{xi}}}\;\}X-\mathrm{effect}\;\;+\sum_{x}\sum_{i}\Delta {\;c}_{i\left(x\right)}\;\bar{c_{x}}\;\bar{s_{\mathrm{xi}}}\;\}I-\mathrm{effect}\;\;+\frac{1}{2}\sum_{x}\sum_{i}\;\left({\;c}_{i\left(x\right)}^{Z}{\;c}_{x}^{Y}-{\;c}_{i\left(x\right)}^{Y}c_{x}^{Z}+{\;c}_{x\left(i\right)}^{Z}c_{i}^{Y}-{\;c}_{x\left(i\right)}^{Y}c_{i}^{Z}\right)\bar{s_{\mathrm{xi}}}\;\}\;X:I\;\mathrm{Interaction}.\;$$

We used the Kitagawa method to decompose differences in crude survival probabilities (S) between two populations, by an age-at-diagnosis composition effect (X-effect), stage-at-diagnosis composition effect (I-effect), age–stage-specific survival effect (survival effect), and an interaction term between the age and stage compositions, such that:(5)$$S^{Y}-S^{Z}=\mathrm{Survival}\;\mathrm{effect}+\mathrm{Age}\;\mathrm{effect}+\mathrm{Stage}\;\mathrm{effect}+\mathrm{Age}:\mathrm{Stage}\;\mathrm{Interaction}.$$

Confidence intervals (CI) for each contribution and for the total difference were calculated using bootstrapping methods. The original sample was randomly resampled with replacement 1000 times, with the sample size kept constant. The decomposition method was then reapplied to each sample and 95% confidence intervals were calculated based on 2.5% and 97.5% percentiles, as similarly suggested by Wang et al. (2000) [18].

### 2.1. Sub-Decompositions to Assess the Effect of the Background Population

The background population can inform about the cancer patient’s chance of survival. For example, a low background survival indicates high risk of death irrespective of cancer and can inform on higher competing risks from other causes. The background age composition is also informative, as an older background population is more likely to have older cancer patients. Therefore, we introduce an extension of the Kitagawa decomposition to quantify the contribution of (1) the background survival and (2) the background age composition to the difference in crude cancer survival between two populations. These contributions can approximate the quantity of the survival effect and age effect that is characteristic of the background population in addition to the differences in survival and age-composition of the cancer patients.

#### 2.1.1. Sub-Decomposition of the Survival Effect

Relative survival (r), i.e., the survival of cancer patients after adjustment for other causes of death, is often a favored measure over crude survival in cancer research. Relative survival is the ratio of the survival observed among cancer patients to the survival observed in a background population (b) with similar demographic characteristics (year, sex, age) ($r_{\mathrm{xi}}=s_{\mathrm{xi}}/b_{x}$), thus controlling for the effect of overall survival.

The survival effect (sE) can be divided into a relative survival effect (rsE) and a background survival effect (bsE), such that sE= rsE+ bsE. Given the formula of $r_{\mathrm{xi}}$, the age–stage-specific survival can be expressed as the product of the relative survival and background survival ($s_{\mathrm{xi}}=r_{\mathrm{xi}}{\;b}_{x}$). Because $s_{\mathrm{xi}}$ can be expressed as a product, the Kitagawa decomposition can be applied to decompose $\Delta \;s_{\mathrm{xi}}$, such that $\Delta \;s_{\mathrm{xi}}=\;\Delta {\;r}_{\mathrm{xi}}\;\bar{b_{x}}+\;\Delta {\;b}_{x}\;\bar{r_{\mathrm{xi}}}$. By replacing $\Delta {\;s}_{\mathrm{xi}}$ in Equation (4) by the previous formulas, we obtain:(6)$$\mathrm{rsE}=\;\sum_{x}\sum_{i}\Delta {\;r}_{\mathrm{xi}}\;\bar{b_{x}}\;\bar{c_{\mathrm{xi}}}$$
(7)$$\;\mathrm{bsE}=\;\sum_{x}\sum_{i}\Delta {\;b}_{x}\;\bar{r_{\mathrm{xi}}}\;\bar{c_{\mathrm{xi}}}\;$$

#### 2.1.2. Sub-Decomposition of the Age Effect

As with survival, the age-at-diagnosis composition is influenced by the age structure of the background population and by the age structure of the cancer patients. Decomposing differences in a composition is however more complex (see the detailed decomposition in Appendix A). Using compositional data analysis (CoDA) techniques [28,29], we can find the age composition $c_{x\left(i\right)}$ as if the background age composition and the relative age composition were equal between populations Y and Z. The procedure presented in Appendix A is not an exact decomposition but approximates the difference in age composition closely: xE≈ bxE+ rxE, where bxE and rxE are the effect of the background age composition and the effect of the relative age composition, respectively.

Confidence intervals for the background effects (age and survival) using the above described method cannot be estimated, as these effects are based on aggregated data only (see Data section below). These contributions should, thus, be seen as an indication of where the differences between populations emerge: From the background structure and survival, or from characteristics of the cancer patients. However, confidence intervals can be estimated for the relative effects (age and survival)

### 2.2. Decomposition of standardized survival

Standardized survival can also be decomposed. Standardized probabilities were obtained by separating survival from the confounding effects of the age compositions and background survival. The average background survival and age composition of the two regions compared was used as “reference population” for the standardization (see Appendix B for details).

## 3. Data

### 3.1. Database

We used data from the Danish Cancer Registry (DCR), in which each tumor is recorded in detail, including histological examination and patient survival. As the Danish five-region classification started in 2007, we used data from 2008 through 2015, thus reflecting the contemporary situation in Denmark. Additionally, the DCR has been using a modernized system since reporting became electronic in January 2008, ensuring more consistent reporting. Over the selected period, the DCR uses the TNM classification to stage cancer and ICD-10 classification of causes of deaths, limiting discontinuities in time series and easing comparisons [30]. Hence, we selected females diagnosed with breast cancer between 2008 and 2010 and calculated the one-year and five-year survival probability. The one-year survival of patient diagnosed between 2011 and 2014 was also studied. Breast cancer was selected because this site had more complete stage-at-diagnosis data than other cancer sites, while being a common cancer.

Within the DCR, 89% of the tumors were morphologically verified, which represents a good validity of the registry [30]. However, information on staging is sometimes missing. The level of completeness depends on the cancer site [31]. For breast cancer diagnosed in Denmark between 2008 and 2010, 5% of the tumors had missing information on tumor size (T), 9% on lymph nodes (N) and 10% on distant metastasis (M).

To obtain information on the background mortality by age, sex and region, we used data from Statistics Denmark [32,33], only available in aggregated form.

### 3.2. Exclusions

The study was performed on malignant neoplasms stated to be primary only; tumors stated as benign, in-situ, of uncertain behavior or secondary were excluded. Additionally, cancer registered from the death certificate or during autopsy only was excluded as the date of diagnosis is the same as the time of death, thus providing no information on survival. Patients living in Greenland or with unknown (or changed) sex or vital status were also excluded, as were cases where the date of censoring occurred before the date of the diagnosis (e.g., when the patient is reported as being departed from Denmark). If individuals were diagnosed with more than one breast tumor (duplicates), a tumor record was kept for the analysis if the patient had no diagnosis of breast cancer five years prior to the diagnosis of interest (between 2008 and 2010 or between 2011 and 2014). In total, 34,723 tumors were kept for the analysis (Table 1).

### 3.3. Staging and Missing Data

We used the TNM data and converted them to prognostic groups using the guide from the American Joint Committee on Cancer (AJCC) 7th edition. In 11% of cases, we were unable to attribute a stage due to at least one missing piece of information on tumor size (T), lymph nodes (N), and/or distant metastasis (M) for women diagnosed with cancer between 2008 and 2010. To avoid bias and information loss, we used multiple imputation to handle missing data. We applied a multiple imputation method using chained equations with the *mice* R package [34,35] to impute a value to the T, N or M missing data, using 15 imputed datasets and 10 iterations. The procedure is detailed in Appendix C.

The decomposition method presented in Section 2 was calculated for each imputed dataset, including the CI procedure. The CIs shown in the following sections were based on 1000 resamples of each of the 15 imputed datasets, thus accounting for the uncertainty of the multiple imputation procedure.

## 4. Results

Central Denmark had the highest survival among the regions for both one-year and five-year survival (95.90% and 82.75% respectively, Table 2) and, thus, served as benchmark for the other regions.

### 4.1. Decomposition of Crude Survival Probabilities (Zealand Versus Central Denmark)

The absolute difference in breast cancer survival between Zealand and Central Denmark was 1.62 percent points for one-year survival and 4.14 for five-year survival, for the period 2008–2010. If Zealand had had the same survival as Central Denmark, the number of breast cancer deaths one year after diagnosis would have been 28.3% lower (corresponding to 40 deaths for the period); five years after diagnosis this number would have been 19.4% (corresponding to 103 deaths for the period).

The survival effect accounted for 31.4% and 43.9% of the difference for the one-year and five-year survival, respectively (Table 3). The survival effect was, however, not significant for the one-year survival. Around 34% of the survival effect can be explained by the background survival components. Central Denmark had better survival from all causes of death combined than Zealand, meaning that cancer patients also benefited from a reduced risk from other causes of death. Female life expectancy for the period 2008–2010 was 81.8 years in Central Denmark and 80.5 in Zealand.

The age effect widened the difference in survival between Zealand and Central Denmark, with a significant contribution for the five-year survival. This difference in the age-at-diagnosis composition of the cancer patients was mainly due the relative age component. The background age component was negative, meaning that Central Denmark had an older background age composition than Zealand. Negative contributions are interpreted as an advantage for Zealand in Table 3.

Zealand also had a more adverse stage-at-diagnosis distribution than Central Denmark, with 44.9% (one-year) and 26.5% (five-year) of the difference in breast cancer survival between the two regions being attributable to the stage effect (both significant, decompositions for other regions in Appendix D).

### 4.2. Decomposition of Standardized Survival Probabilities

For survival standardized by background survival and age, the differences in the one-year and five-year survival between the Zealand and Central Denmark was 1.24 and 2.39 percent points, respectively, for the period 2008–2010. Table 4 shows the decomposition of the standardized survival by stage-specific survival and stage composition, using the Kitagawa method with one composition effect (Equation (2)). Compared with the contributions of relative survival and stage from the decomposition presented in Table 3, the stage and relative survival contributions remained equal at a two-decimal rounding when decomposing the crude and standardized survival probabilities. However, after standardizing by age, we cannot separate out the interaction effects, which is contained within the stage effect. Standardizing does not affect the absolute contributions of the components, when using the average between populations as reference (Appendix B). After removing the age and background effects, the stage effect is the dominant contributor to the difference in the one-year survival between the two regions, explaining 73.5% of the difference.

Figure 1 shows the standardized survival decomposition between Central Denmark and the four other regions. After standardization, Southern Denmark has a better survival than Central Denmark, which is mainly attributed to the survival effect. Central Denmark had a better stage-specific survival than the other three regions. The survival effect contributed to 79.8% and 86.0% of the difference in breast cancer one-year and five-year survival, respectively, between Northern Denmark and Central Denmark.

The stage-effect contributed to the disadvantage of Zealand, but played in favor of the Capital region for the five-year survival. There was no significant stage effect explaining the difference between Central Denmark and the Southern and the Northern regions.

### 4.3. Diminishing Differences over Time

The five-year survival cannot be calculated for more recent years, but the one-year survival can be calculated for patients diagnosed with breast cancer between 2011 and 2014. Differences between Central Denmark and the other regions have decreased between the periods 2008–2010 and 2011–2014 (Figure 2), showing evidence of progress towards equality. The survival effect was significant for three regions in 2008–2010, but only for Northern Denmark in 2011–2014, for which it decreased four-fold over the study period.

Central Denmark had an advantage over Southern Denmark in the most recent period, but it is small (0.44 percent points), and the contributions are not significant.

The stage effect was still significant for Zealand in the most recent period, but smaller, and explains 84.6% of the difference in breast cancer survival with Central Denmark in 2011–2014.

## 5. Comparison with the Cox Proportional Hazard Model

The Kitagawa decomposition differs from the (commonly-used) Cox proportional hazard (CPH) model, and other types of regression models, in important ways (Table 5).

First, the CPH model assesses which variables influence survival. For example, an increase in the age and stage at diagnosis increases the hazard (Table 6) and decreases survival. In contrast, the decomposition method quantifies contributions of specific variables to the difference in survival between two populations. 

Second, the CPH, and other forms of regression model, estimate coefficients. The coefficients act multiplicatively on the variables’ value and are used to predict survival for an individual with specific characteristics. The decomposition produces variable-specific contributions to the difference, summing up to the total survival difference. The contributions are generally an aggregated value for each variable, without distinction for the value of the variable (e.g., stages 1 to 4).

Third, the CPH estimates relative differences between values of a variable. For example, the ratio of the hazard functions of Zealand/Central Denmark is higher than 1 (Table 6), meaning that people diagnosed with breast cancer in Zealand had a higher hazard than people with similar characteristics in Central Denmark. This approach does not inform, however, on why this difference between region occured. In contrast, the decomposition approach uses absolute differences between two populations. The use of absolute rather than relative differences allows one to quantify the survival that could be gained by improving, for example, stage at diagnosis to the level of a reference region or population: If Zealand had the same stage at diagnosis distribution as Central Denmark in 2008-2010, the five-year crude survival probability (Table 3) would have been 79.80% rather than 78.61%. It also allows one to quantify directly the number of deaths that could be avoided if one of the components were to change. For example, giving Zealand the stage-at-diagnosis distribution of Central Denmark reduces the number of deaths due to breast cancer one year after the diagnosis by 16.0% for patients diagnosed between 2008 and 2010 (23 deaths) and by 5.6% five years after the diagnosis (30 deaths). Most articles tend to report relative measures only, but existing recommendations suggest reporting both relative and absolute measures [36].

Fourth, the CPH model makes structural assumptions, primarily the hazards are proportional. With the decomposition model, no distributional or structural assumptions are required: The compositions and component-specific rates or probabilities observed in two populations are directly compared and their effects on survival are quantified.

Finally, the CPH model requires individual data, while the decomposition method can also be used on aggregated data. However, if aggregated data are used, new ways to calculate confidence intervals should be found other than that suggested in the paper.

The CPH and decomposition models serve different purposes and the use of one rather than the other should be determined by the aim of the study. If the aim is to understand the determinants of cancer survival, the CPH model, or other regression models, should be used. However, if the aim is to understand differences in survival between populations or quantify potential gains in survival by modifying one variable, the introduced decomposition method can be preferable. Both methods could also be used to complement each other.

## 6. Discussion

We presented a non-parametric decomposition method that uncovers the causes of differences in cancer survival probabilities between populations. In the test case of the Danish regions, we found that later stage at diagnosis explained a large share of the difference in breast cancer survival between Zealand and Central Denmark, which tentatively suggests that Zealand could improve cancer survival by diagnosing at an earlier stage, in addition to recent and ongoing improvements.

By the end of 2007, all Danish regions were required to start a breast cancer screening program, the rollout being completed in 2009. However, some differences between regions remained. Zealand recorded more fluctuations in the breast cancer detection rates over time (in contrast to the other regions) with a particularly low detection rate of 0.53% compared with the national average of 0.61% in the fourth screening round (2014–2015) [37]. This could explain, in part, the later stages at diagnosis in Zealand and might be caused by a shortage of experienced radiologist in the region [37].

Lower socioeconomic status has been associated with later cancer stage at diagnosis [38]. Zealand has the highest proportion of residents with low education level among the Danish regions [39], which could also explain its more adverse stage-at-diagnosis distribution. However, Ibfelt et al. (2018) [17] found that even after controlling for differences in socioeconomic status (education and income), the odds ratio of being diagnosed at a later stage remained higher in Zealand than in the Capital region for malignant melanoma. This led the authors to suspect differences in the referral process to specialized care between regions. Other possible explanations include fewer specialized doctors in the outer regions, such as Zealand and Northern Denmark, and other unmeasured social, cultural and behavioral factors [17]. Patient awareness of breast cancer symptoms is high in Denmark, especially in highly educated respondents [40], which suggests that patient delay may be a factor in regions where education is generally lower.

Given the importance of family doctors in the Danish healthcare system, without whose referral one cannot consult a specialist, regional differences in organization, attitudes and number of GPs are likely to lead to some differences in stage distribution. Looking into regional differences in England, Maclean et al. (2015) [41] found that for female breast cancer, being in a practice with short waiting times until referral or detection was associated with a lower proportion of patients diagnosed in stage 3 or 4 rather than stage 1 or 2. Membership of a practice where people thought it less easy to book an appointment was associated with a higher percentage diagnosed later. It would be helpful to know how this translates to the case of Zealand GPs. Presumably, such an effect will depend on the quality, attitudes and organization of family doctors, which may lead to regional differences.

As for stage-specific survival, there are known cases where, while treatment was in principle available, the application of active treatment was different between regions in one country, such as lung cancer in England [42]. However, the Danish Breast Cancer Group (DBCG) established mandatory treatment guidelines, especially regarding the surgical treatment and the (neo)adjuvant treatment [43] (DBCG guidelines, www.DBCG.dk). In principle, patients should get the same treatment across all Denmark’s hospitals. Furthermore, DBCG regularly publish a quality indicator report to identify any variation in the treatment of early breast cancer, showing only small differences [43] (DBCG quality indicator report, www.DBCG.dk). Differences in treatment are, thus, unlikely to cause the differences observed in stage-specific survival between Danish regions (survival effect).

It has also been found that rural dwellers have poorer cancer survival [44]. It is unclear why this would apply only to Northern Denmark, although the vicinity of the capital may make Zealand effectively a little less “rural” than Northern Denmark.

In 2007, many changes occurred regarding breast cancer diagnosis and treatment in Denmark, including the screening program, updates of the national guidelines, and the new regional scheme. These changes might be the cause for the decreasing differences across regions over time, but we cannot assess if or which of these changes are responsible for the convergence, or if it can be the result of previous programs.

More information could be added in the decomposition to further explain the regional differences. If additional data were available, possible extensions could include the compositional difference of socioeconomic status, smoking habits, or medical treatments. The contributions of unspecified components are grouped in the survival-effect, or in the composition effects if an unspecified composition correlates with the age or stage compositions. For example, in Table 3, if the stage at diagnosis was not included in the analysis, the survival effect would be approximated by the sum of the survival and stage effects (e.g., 1.24 for the one-year survival) and the age effect would be the sum of the age and interaction effects (e.g., 0.38 for the one-year survival). This is similar to unmeasured confounders in regression analysis.

## 7. Conclusions

This paper illustrates the utility of adopting and extending the Kitagawa decomposition to cancer research. The method allows us to understand differences in survival between populations by quantifying the exact contributions of specific variables to this difference. Such quantification can help policy makers and health care professionals improve overall cancer survival, tuning their actions to the dominant contributions. The method presents some advantages compared with other models commonly used in survival analysis, such as the Cox proportional hazard model, when it comes to understanding differences between populations. We argue that decomposition methods are valuable tools and provide a powerful complementary analysis for cancer and public health research.

## Figures and Tables

**Figure 1 ijerph-16-03093-f001:**
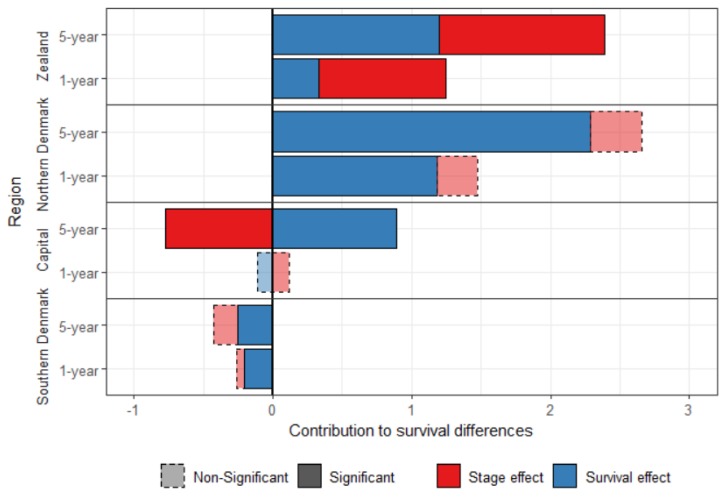
Decomposition of the one-year and five-year standardized survival difference between Central Denmark and the other Danish regions, for women diagnosed with breast cancer between 2008 and 2010.

**Figure 2 ijerph-16-03093-f002:**
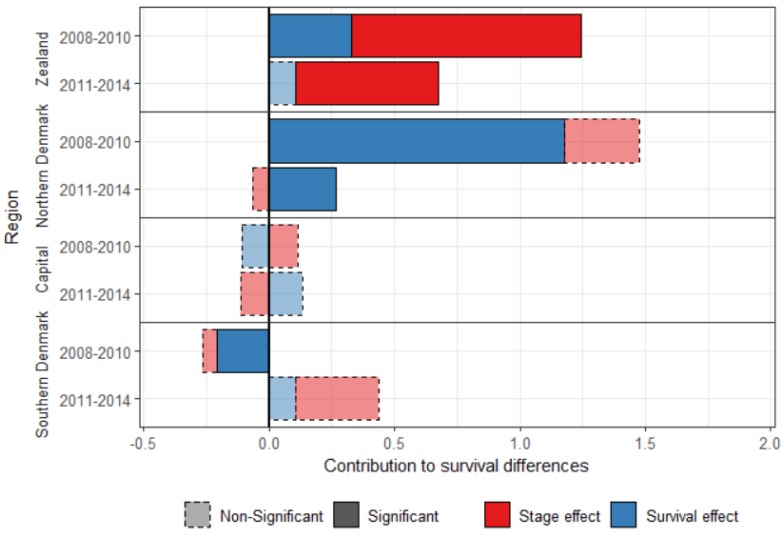
Decomposition of the one-year standardized survival difference between Central Denmark and the other Danish regions, for women diagnosed with breast cancer between 2008–2010 and between 2011–2014.

**Table 1 ijerph-16-03093-t001:** Number of diagnoses of primary malignant neoplasms from breast cancer for women diagnosed between 2008 and 2010 and between 2011 and 2014 in Denmark.

Period	Eligible	Excluded			Included
		Duplicates	Death Certificate Only	Other	
2008–2010	16,019	90 (0.6%)	83 (0.4%)	34 (0.2%)	15,812 (98.7%)
2011–2014	19,176	85 (0.4%)	103 (0.4%)	77 (0.4%)	18,911 (98.6%)

**Table 2 ijerph-16-03093-t002:** One-year and five-year crude survival probabilities (as percentages) for the five Danish regions, for women diagnosed with breast cancer between 2008 and 2010, ordered by increasing five-year survival.

Region	One-Year Survival (%)	Five-Year Survival (%)
Zealand	94.28	78.61
Northern Denmark	94.24	78.86
Capital region	95.37	80.45
Southern Denmark	95.85	82.34
Central Denmark	95.90	82.75

**Table 3 ijerph-16-03093-t003:** Decomposition of the difference in breast cancer one-year and five-year crude survival. between Zealand and Central Denmark, 2008–2010.

Components	Contributions	CI (95%)	%	Sub-Components	Contributions	%
**One-year survival**						
Survival	0.51	(−0.44, 1.63)	31.4	Relative survival	0.33	65.3
				Background survival	0.18	34.7
Age	0.19	(−0.26, 0.55)	12.0	Relative age	0.25	128.9
				Background age	−0.06	−28.9
Stage	0.73	(0.14, 1.19)	44.9	-	-	-
Age–stage interaction	0.19	(−0.08, 0.52)	11.7	-	-	-
Total	1.62	(0.54, 2.73)	100.0			
**Five-year survival**						
Survival	1.82	(0.09, 3.62)	43.9	Relative survival	1.20	65.8
				Background survival	0.62	34.2
Age	1.13	(0.31, 1.92)	27.4	Relative age	1.28	112.6
				Background age	−0.14	−12.6
Stage	1.10	(0.25, 1.94)	26.5	-	-	-
Age–stage interaction	0.09	(−0.32, 0.50)	2.2	-	-	-
Total	4.14	(1.93, 6.29)	100.0			

**Table 4 ijerph-16-03093-t004:** Decomposition of the difference in breast cancer one-year and five-year survival standardized (by age and background survival) between Zealand than in Central Denmark, 2008–2010.

Standardized Survival				Crude Survival (Table 3)	
Components	Contributions	CI (95%)	%	Components	Contributions
**One-year survival**					
Survival	0.33	(0.12, 0.58)	26.5	Relative Survival	0.33
Stage	0.92	(0.43, 1.36)	73.5	Stage + Interaction	0.92
Total	1.24	(0.97, 1.43)	100.0	Sum	1.24
**Five-year survival**					
Survival	1.20	(0.84–1.45)	50.1	Relative Survival	1.20
Stage	1.19	(0.32–2.05)	49.9	Stage + Interaction	1.19
Total	2.39	(2.02–2.62)	100.0	Sum	2.39

**Table 5 ijerph-16-03093-t005:** Summary of the Cox proportional hazard and Kitagawa decomposition models characteristics.

	Cox Proportional Hazard	Kitagawa Decomposition
What is measured?	Determinants of survival	Determinants of survival differences
Model outputs	Coefficients	Contributions
Difference measured	Relative	Absolute
Key assumption	Proportional hazards	None
Data	Individuals	Individuals and aggregates

**Table 6 ijerph-16-03093-t006:** Results from a Cox proportional hazard model, comparing hazards from Central Denmark and Zealand, 2008–2010.

Variables	Coefficient	Exp (Coefficient)	CI (95%)
Age at diagnosis	0.11	1.12	1.10, 1.14
Stage at diagnosis	2.24	9.38	6.40, 13.75
Region Zealand	1.10	2.99	1.35, 6.65
Age: stage	−0.02	0.98	0.98, 0.99
Age: region	−0.01	0.99	0.98, 1.00
Stage: region	−0.18	0.83	0.74, 0.94

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
