# Peer review of "Understanding Differences in Cancer Survival between Populations: A New Approach and Application to Breast Cancer Survival Differentials between Danish Regions"

_ijerph, 2019, doi:10.3390/ijerph16173093_

Round 1

Reviewer 1 Report

IJERPH-558145-July-24-2019

This study describes a novel extension of the standard Kitagawa decomposition method to allow for sub-decompositions of background survival probability and age composition, with a major strength in addressing the confounding effect of background population. Other strengths a thorough discussion of possible reasons for differences in later cancer stage at diagnosis and stage-specific survival observed in the study population. The proposed method is applied to quantify the differences in 1-year and 5-year breast cancer survival probabilities between populations in five Danish regions. Contributions of differences in stage at diagnosis, population age structure and stage-age-specific survival probability are investigated. The most significant difference in stage at diagnosis exists between Zealand and Central Denmark, indicating that action is required for Zealand to ensure timely diagnosis of breast cancer. Moreover, longitudinal comparison of 1-year survival probability shows diminishing gaps between Central Denmark and other regions, indicating efforts towards equality. I have the following critiques, that if addressed, would strengthen the manuscript.

Major

Figure 2 – provide explanation on the change in direction of contribution for Southern Denmark. Compared to the referenced Central Denmark region, it is interesting to see that the survival of South Denmark was better between 2008 and 2010, whereas worsened between 2011 and 2014. An explanation of this situation would help with the interpretation. Demonstrate how the proposed method can be used as a complementary analysis for a regression model. Although mentioned in lines 332-333 that more information could be added to the decomposition, it is possible that the decomposed effects become very small as the number of factors increase, in which case relative differences measured from regression models might be more informative. Thus there are some situations in which one method but not the other would be useful and other situations where both methods could be used together. If authors do not have the space or time to fit regression models and compare findings and results between those models and decomposition, please describe this as a limitation and include text in the discussion section to help readers think through situations wherein decomposition and/or regression would be a preferred approach.

Minor

Lines 233 - rephrase “Around 34% of the survival effect can be explained by the background survival components”. This statement is appropriate when both components contribute positive effects. However, (-28.9%) of the age effect explained by background age component seems inappropriate. The last column (Table 3) provides similar information as does column 6, thus could be excluded from the table. A possible rephrase could be based on the data in column 6. It is more important to explain why the age effect contributed by its sub-components can have opposite signs. Explain why treatment of breast cancer is not considered as a composition to the survival differences in the methods and/or limitation sections. Line 121 - explain what the distribution means. Notation in equations (4), (6) and (7) – explain the subscript . Typos Line 121 - explain what the i distribution means. Line 525 - “we think this is procedure is justified”. Line 533, typo – appendix C. Line 549, typo – appendix D.

Author Response

We would like to thank the reviewer for his\her useful comments. Please find below our answer to each comment.

Figure 2 – provide explanation on the change in direction of contribution for Southern Denmark. Compared to the referenced Central Denmark region, it is interesting to see that the survival of South Denmark was better between 2008 and 2010, whereas worsened between 2011 and 2014. An explanation of this situation would help with the interpretation.

Re: We now highlight this change of direction on lines 285-286.

Even if the direction of the contributions changed, the contributions in 2011-2014 are not significant. We thus have little indication of why this happened. It might simply be due to fluctuations or a small number of diagnoses.

Demonstrate how the proposed method can be used as a complementary analysis for a regression model. Although mentioned in lines 332-333 that more information could be added to the decomposition, it is possible that the decomposed effects become very small as the number of factors increase, in which case relative differences measured from regression models might be more informative. Thus there are some situations in which one method but not the other would be useful and other situations where both methods could be used together. If authors do not have the space or time to fit regression models and compare findings and results between those models and decomposition, please describe this as a limitation and include text in the discussion section to help readers think through situations wherein decomposition and/or regression would be a preferred approach.

Re: We added a section (Section 5) discussing the differences between our model and a Cox proportional hazard model.

If a variable explains most of the difference in survival, e.g. stage, adding more variables to the decomposition will only divide the survival effect and the stage effect should remain, unless it correlates strongly with all newly included variables (lines 399-405).

Lines 233 - rephrase “Around 34% of the survival effect can be explained by the background survival components”. This statement is appropriate when both components contribute positive effects. However, (-28.9%) of the age effect explained by background age component seems inappropriate.  The last column (Table 3) provides similar information as does column 6, thus could be excluded from the table. A possible rephrase could be based on the data in column 6. It is more important to explain why the age effect contributed by its sub-components can have opposite signs.

Re: We added a discussion on the age contributions to the difference and an interpretation of negative contributions on lines 240-244.

Explain why treatment of breast cancer is not considered as a composition to the survival differences in the methods and/or limitation sections.

Re: We did not have this information in our dataset (line 398).

Line 121 - explain what the distribution means, Typos Line 121 - explain what the i distribution means.

Re: x and i can be any compositions. Here, we defined x as age and i as stage. But researchers can change these variables. We explain this more clearly in lines 120 and 122-123 and we defined x and i in lines 126-129.

Notation in equations (4), (6) and (7) – explain the subscript .

Re: The subscripts have the same definition as in equation 1 to 3. The dot on the i disappears under the bar when using the Word equations. We hope the editorial office can help us fix this issue.

Line 525 - “we think this is procedure is justified”.

Re: The typo has been corrected (line 585).

Line 533, typo – appendix C.

Re: The typo has been corrected (line 594).

Line 549, typo – appendix D.

Re: The typo has been corrected (line 610).

Reviewer 2 Report

- Line 34 there is a typo …. Country[es   should be “countries”

- Should Zealand be New Zealand

- The authors should compare the 1 - and 5-year survival outcome values in their

cohort using standard methods (e.g.   HR, CI, regression models, etc…) with their

decomposition method to prove their method estimates/performs better.   

- The authors should provide figures and/or tables comparing the values between  

method types.  

- The way the data is shown in the figures is confusing as stage of breast cancer

appears to be grouped together.  The authors should perform the model stratified by

stage and note potential causes for the differences observed in the discussion.

   - Again, in line 293, the authors state that “ the later stages at diagnosis in Zealand….”  But the data appears to combine all of the stages of breast cancer together.  Are only stages 3 and 4 breast cancer patients included in the cohort? 

Author Response

We would like to thank the reviewer for his\her useful comments. Please find below our answer for each comment.

- Line 34 there is a typo …. Country[es   should be “countries”

Re: The typo has been corrected (line 34).

- Should Zealand be New Zealand

Re: Zealand is the official English name of a region in Denmark. It is the region surrounding the Capital.

- The authors should compare the 1 - and 5-year survival outcome values in their cohort using standard methods (e.g.   HR, CI, regression models, etc…) with their decomposition method to prove their method estimates/performs better.   

- The authors should provide figures and/or tables comparing the values between method types.  

Re: We added a section (Section 5) comparing the introduced method with the Cox proportional hazard model. However, due to the different nature of the models, a direct comparison between coefficients can hardly be done. We explain these differences in more detail in the revised manuscript.

- The way the data is shown in the figures is confusing as stage of breast cancer appears to be grouped together.  The authors should perform the model stratified by stage and note potential causes for the differences observed in the discussion.

- Again, in line 293, the authors state that “ the later stages at diagnosis in Zealand….”  But the data appears to combine all of the stages of breast cancer together.  Are only stages 3 and 4 breast cancer patients included in the cohort? 

Re: Unlike regression models, the introduced method compares distributions, i.e. we compare the proportions of cancer diagnosed in each stage in both populations. If population A has a stage distribution from 1 to 4 equal to (0.45,0.30,0.20,0.05) and population B (0.40,0.28,0.22,0.10), then most likely this population A will have higher survival. We thus assess how much of this difference in distributions is responsible for differences in the overall survival. This has been detailed in the new section (lines 311-313).  

Reviewer 3 Report

This is a non-standard way of survival analysis. The approach makes sense to me (I am a statistician).  Because it is not the conventional way, the authors better compare the result with the one from the Cox model (more like a gold standard). In fact, it is not that difficult to find the comparable results from the two approaches (i.e., the current approach and Cox model). I think the author should compare these two methods in order to "sell" this approach. Another direction of the manuscript should work on is to tell us when we should use or should not use this approach, in comparison to Cox model (this is also distribution-free approach).

Overall, I think this is an alternative way of the survival analysis and it is helpful in a different way (as an alternative).  

Author Response

We would like to thank the reviewer for his\her useful comments. We added a section (Section 5) discussing the difference between the Cox model and the introduced decomposition method, including a discussion of when the different approach should be used.

Round 2

Reviewer 1 Report

The authors have responded to all my concerns and I think this manuscript should be accepted for publication.

Reviewer 2 Report

The authors have adequately addressed my comments and concerns.

Reviewer 3 Report

Thank you for your response. All my concerns have been addressed.